



# Qualitative yaw stability analysis of free-yawing downwind turbines

Gesine Wanke[1], Morten H. Hansen[2], and Torben J. Larsen[3]

[1]Suzlon Blade Science Center, Havneparken 1, 7100 Vejle, Denmark
[2]Mads Clausen Institute, University of Southern Denmark, Alison 2, 6400 Sønderborg, Denmark
[3]DTU Wind Energy, Technical University of Denmark, Frederiksborgvej 399, 4000 Roskilde, Denmark

**Correspondence:** Gesine Wanke (gesine.wanke@suzlon.com)

**Abstract.** This article shows qualitatively the yaw stability of a free yawing downwind turbine and the ability of the turbine to align passively with the wind direction, using a two degree of freedom model. An existing model of a Suzlon S111 upwind 2.1 MW turbine is converted into a downwind configuration with a 5° tilt and a 3.5° downwind cone angle. The analysis shows that the static tilt angle causes a wind speed dependent yaw misalignment of up to -19° due to the projection of the torque onto
the yaw bearing and the skewed aerodynamic forces caused by wind speed projection. With increased cone angles, the yaw stiffness can be increased for better yaw alignment and the stabilization of the free yaw motion. The shaft length influences the yaw alignment only for high wind speeds and cannot significantly contribute to the damping of the free yaw mode within the investigated range. Asymmetric flapwise blade flexibility is seen to significantly decrease the damping of the free yaw mode, leading to instability at wind speeds higher than 19 ms$^{-1}$. It is shown that this additional degree of freedom is needed to predict
the qualitative yaw behaviour of a free yawing downwind wind turbine.

## 1 Introduction

With the increase in wind turbine rotor size and the increase in rotor blade flexibility, downwind concepts where the rotor is placed behind the tower re-experience an increase in research effort. The downwind concept potentially comes with the option of a passive yaw alignment. A passive yaw concept could save costs on the yaw system, decrease the maintenance and reduce
the complexity of the yaw system. In situations where one side of a rotor under yawed inflow is higher loaded than the other, the resulting forces on the blades create a restorative yaw moment and could potentially align the rotor with the wind direction. These passive yaw systems have been investigated already in the 1980s and the early 1990s. Corrigan and Viterna (1982) studied the free yaw performance of the two bladed, stall controlled MOD-0 100 kW turbine with different blade sets. They observed that the turbine aligns with the wind direction at yaw errors between -45° and -55°. The yaw motion was positively
damped for short term wind variations at these positions. The power production was significantly lower compared to the forced yaw alignment. An improvement of the alignment with the wind direction could be achieved by the elimination of the turbine tilt. Wind shear on the other hand was observed to have a negative influence on the yaw alignment.

In further tests on the MOD-0 100 kW turbine, Glasgow and Corrigan (1983) investigated the influence of bend-twist coupling and the airfoil at the tip section for a tip-controlled configuration. Their study showed, a strong dependency of the yaw align-
ment on the wind speed. For the bend-twist coupled rotor the minimum yaw error was observed to be -25°. The comparison



between two different tip airfoils showed that the alignment could be significantly improved with an airfoil with favourable characteristics.

Olorunsola (1986) investigated the yaw torque for different yaw inflow angles. He emphasized the risk of stall induces vibrations and increased fatigue loads in cases where the aerodynamically provided yaw torque cannot overcome the frictional

torque of the yaw bearing, leading potentially to an operation with high yaw misalignment.

In 1986, the University of Utah and the Solar Energy Research Institute in the US started to develop and validate a model for the prediction and understanding of yaw behaviour. In a time domain modelling approach, they coupled the flapwise blade motion to the yaw motion. In several studies (e.g Hansen and Cui (1989), Hansen et al. (1990) and Hansen (1992)), the resulting YawDyn tool was used to reproduce the results of measurement campaigns and to identify the most influential parameter on

the free yaw behaviour. The researchers emphasized the importance to include dynamic stall effects and skewed inflow model in the prediction of yaw behaviour. They could further show the influence of blade mass imbalances, tower shadow, rotor tilt and horizontal and vertical wind shear as the contribution to asymmetry of the rotor loading from flapwise blade root bending moments. While the study showed that the yaw behaviour could be simulated qualitatively, the tool was not able to capture the quantitative yaw dynamics correctly in all test cases.

Other modelling approaches were chosen for example by Madsen and McNerney (1991) who developed a frequency domain model to study the statistics of yaw response and power production of a 100 kW turbine in dependency of a turbulent wind regime. They confirmed that the horizontal wind shear is a major source for yaw errors and the related power loss.

Pesmajoglou and Graham (2000) on the other hand predicted the yaw moment coefficient for different sized turbine models with a free vortex lattice model. They showed a good agreement of the mean yaw moment with wind tunnel experiments in

cases where airfoil stall does not show a large contribution to the yaw moments. In these cases, their model could successfully predict the variation of the yaw moment coefficient in a turbulent wind field.

Verelst and Larsen (2010) investigated the restoring yaw moment due to yawed inflow on a stall regulated 140 kW machine with stiff rotor blades and different cone configurations. They showed an increase of the restorative effect on the yaw moment from higher cone angles, because the cone angle increases the imbalance of the rotor forces and therefore the restorative yaw

moment. However, for negative yaw errors they showed that the midspan part of the blades contributes to a decrease in the restorative yaw moment, related to the stall effect at rated wind speed. This effect could be reduced, but not eliminated with the highest tested cone angles.

Picot et al. (2011) studied the effect of swept blades on a coned rotor on a 100 kW stall regulated turbine. They investigated the restorative yaw moment in a fixed yaw configuration, as well as, the yaw alignment in a free yaw configuration. In their

study, they observed yaw oscillations around rated wind speed. The azimuth variation of inflow condition due to wind shear increased the yaw oscillation. They confirmed that the inner part of the blade being in deep stall, contributes to the reduction of the restorative yaw moment. With backward swept blades, the destabilizing effect of the stall was reduced, but occurred over a larger wind range. Sine the blade was passively unloaded at higher wind speeds and the inflow condition due to the position of the blade segments differed along the blade due to deformation, the different blade segments were subject to stall at different

wind speeds.





Kress et al. (2015) used a scaled model of a commercial 2 MW downwind turbine to compare the restorative yaw moment in a water tunnel in a downwind and upwind configuration. They compared the influence of different cone angles, different yaw angles and different tip speed ratios close to optimum tip speed ratio. They observed a restorative yaw moment for all downwind configurations. In the upwind configuration only configurations with high cone angles showed a restorative yaw

moment, which was seen to be significantly smaller than in the downwind configuration.

Verelst et al. (2016) showed measurements of a 280 W downwind turbine in a open jet wind tunnel. They released the rotor yaw from large yaw errors ($\pm 35°$) and measured the angle where the rotor would passively align with the wind direction, as well as the dynamic yaw response. They tested the angle of alignment for a rotor with stiff/flexible blades and swept/non swept blades. They observed that the equilibrium yaw angle was not exactly zero and they assumed that the yaw moment is too small

to overcome the bearing friction and the rotor inertia. They further showed that the steady state yaw angle found from initially negative yaw errors was higher than for positively yaw errors. They stated that the reason could be an asymmetry in the inflow due to the tower shadow or a non zero steady state yaw angle for a zero yaw moment. They further found a different yaw stiffness for positive and negative yaw errors, leading to different system responses with an under-damped response only for positive yaw errors.

In this article the influence of geometrical parameters such as cone, tilt and shaft length on the equilibrium position of a free yawing pitch regulated 2.1 MW downwind turbine are considered. The influence of cone, shaft length, and the center of gravity position of the nacelle on the damping of the yaw mode is further considered from a simple two degree of freedom model with the free yaw and tower side-side motion. It is shown that a full alignment with the wind direction is only achievable without tilt angle of the turbine and inclination angle of the wind field. It is shown that large cone angles increase the alignment with the

wind direction and the damping of the free yaw mode. Finally, it is shown that flapwise blade flexibility needs to be added to the two degree of freedom model, as it will significantly reduce the damping and the yaw equilibrium could become unstable.

## 1.1 Yaw moment, aerodynamic yaw stiffness and damping mechanisms

The total moment on the yaw bearing is determined by different mechanisms creating the yaw loading. The following estimation sketches the main contributors to the total yaw moment $M_{yaw}$.

$$M_{yaw} \approx M_{Q,\delta} + M_{W,\delta} + M_a + M_{W,\theta} + M_{W,\gamma_c,\theta}$$

where the torque projection $M_{Q,\delta}$ and the moment due to wind speed projection from tilt angle $M_{W,\delta}$ are dependent on the tilt angle $\delta$. The moment due to induction variation from the skewed yaw inflow is $M_a$, the moment due to projection of the wind speed with the yaw angle is $M_{W,\theta}$ and the moment due to wind speed projections with a combined cone and yaw angle is $M_{W,\gamma_c,\theta}$.

There are two yaw moment contributions due to the tilt angle. The first one is a projection of the main shaft torque $M_{Q,\delta}$ onto the yaw axis with the sine of the tilt angle (structural effect of tilt). As power production changes with wind speed $W$, the yaw moment due to torque changes with wind speed. In case of a yaw misalignment, the torque is reduced and influences the yaw moment accordingly. The second moment caused by the tilt angle $M_{W,\delta}$ is due to the wind speed projection, illustrated in Fig. 1





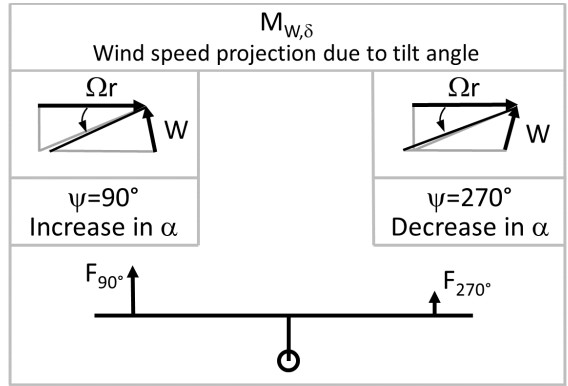
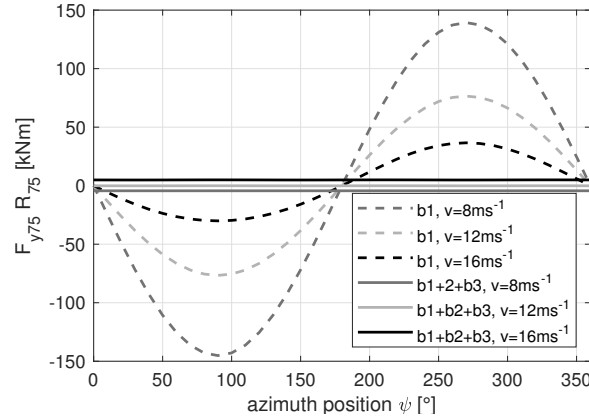

(a) Yaw moment sketch due to wind speed projection with tilt angle.

(b) Yaw moment estimate from $F_y$ at $R = 75\%$ and $\delta$=5°.

**Figure 1.** Aerodynamic yaw moment for the tilt angle of a downwind rotor sketched on the left and the roughly estimated respective variation of yaw moment of the force at 75% rotor radius on the right.

(aerodynamic effect of tilt). Figure 1 (a) shows that the projection of the incoming wind speed is subtracted from the rotational speed $\Omega R$ when the blade moves up (azimuth position of $\psi = 90°$) and added to the rotational speed when the blade moves down ($\psi = 270°$). The difference in projected wind speed creates a variation of angle of attack $\alpha$ over the azimuth position. Figure 1 (b) shows the variation of the yaw moment over azimuth position for different wind speeds due to the force at 75%

of the rotor radius. It can be seen that the sum of the loading from three blades is not zero. In the attempt to isolate the effect of wind speed projection from a tilt angle the interaction with other effects, e.g. a combination of several angle projection (tilt, cone and yaw) or the skewed inflow model for tilted inflow are not included in the figure. The two tilt dependent moments, $M_{Q,\delta}$ and $M_{W,\delta}$ will cause a yaw misalignment for any free yawing turbine with a structural tilt angle. An inclination angle of the wind field would also cause a moment from projections as $M_{W,\delta}$.

The moment due to induction variation over the rotor plane $M_a$, the moment due do wind speed projections from the a yaw angle $M_{W,\theta}$ and the moment due to wind speed projections from a combination of yaw and cone angle $M_{W,\gamma_c,\theta}$ are restorative moments. The restorative moments are creating an aerodynamic yaw stiffness as shown in Fig. 2. A yaw displacement will introduce a variation of induction over the rotor plane, due to the skewed inflow model, as one half of the rotor is positioned deeper in the wake than the other half. The upstream pointing blade is therefore higher loaded and a restoring yaw moment is

created (Fig. 2 (a)). It can be seen in Fig. 2 (b) that relatively high yaw angles are required to create a significant restorative yaw moment from the variation of induction over the rotor plane compared to other stiffness mechanisms.

The yaw displacement creates a projection of the incoming wind speed. When the blade is pointing down ($\psi = 0°$), the projected wind speed component is added to the rotational speed, while it is subtracted to rotational speed, when the blade is pointing up. The resulting variation of angle of attack is the reason for an in-plane force at the hub center that creates a moment

with the arm of the shaft length (Fig. 2 (c)). This effect creates the smallest yaw moment of the discussed effects. However,





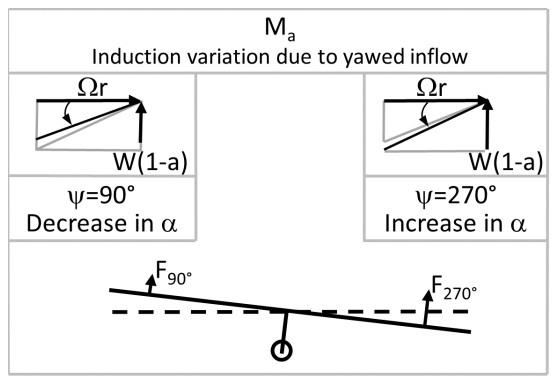

(a) Stiffness mechanism: induction variation.

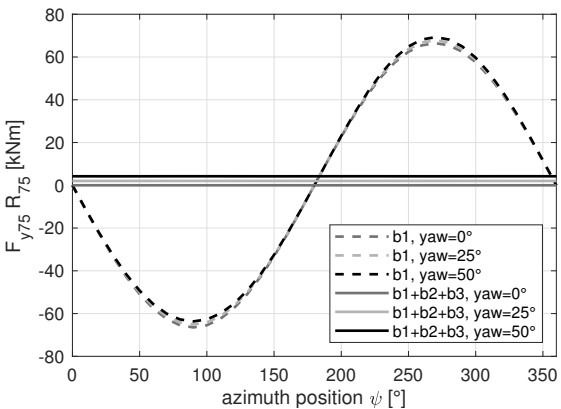

(b) Yaw moment estimate from $F_y$ at $R = 75\%$.

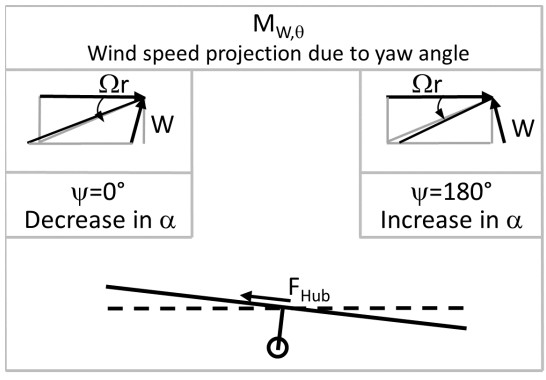

(c) Stiffness mechanism: yaw projection.

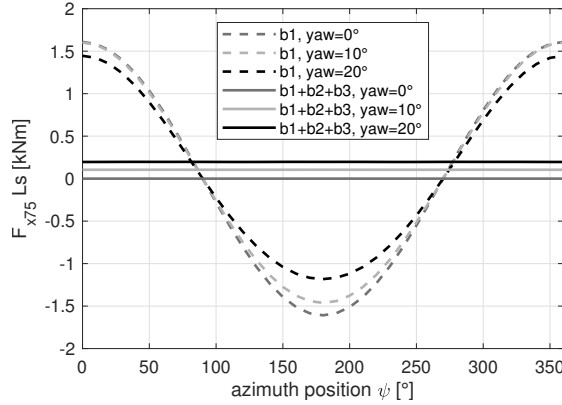

(d) Yaw moment estimate from $F_x$ at $R = 75\%$.

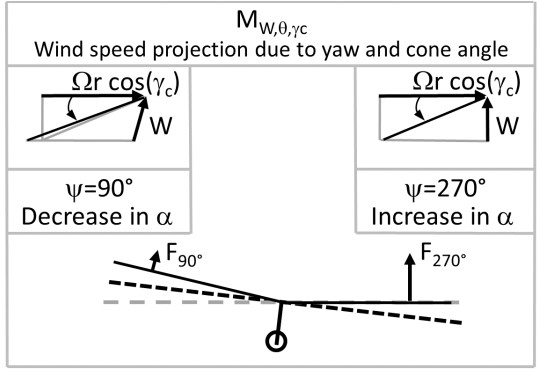

(e) Stiffness mechanism: yaw and cone projection.

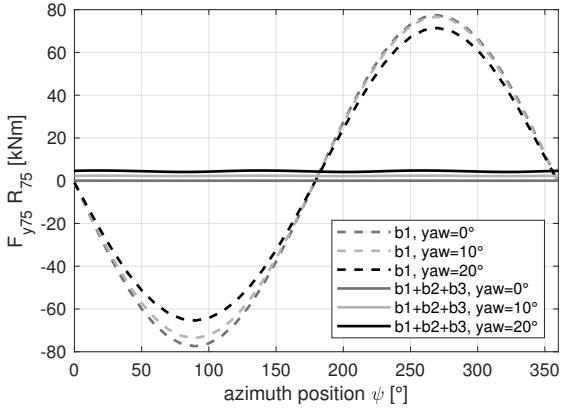

(f) Yaw moment estimate from $F_y$ at $R = 75\%$ and $\gamma_c = 3.5°$.

**Figure 2.** Aerodynamic mechanisms for yaw stiffness of a downwind rotor on the left and the roughly estimated respective variation of yaw moment of the force at 75% rotor radius for wind speeds of 12 ms$^{-1}$ over azimuth position on the right.



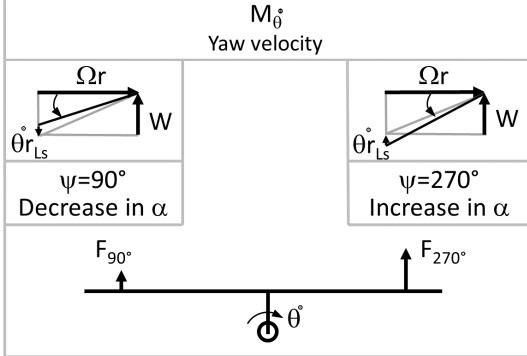

**Figure 3.** Aerodynamic mechanism for yaw damping for a downwind rotor.

with higher pitch angles, the contribution becomes larger at higher wind speeds, due to the flapwise force component that is projected to the in-plane forces.

In the case of coning, there is a difference in the projected wind speed between the left and the right side of the rotor when the rotor is yaw misaligned, resulting in a difference in angle of attack. From the difference in loading, a restoring yaw moment is
created (Fig. 2 (e)). It can be seen in Fig. 2 (f) that relatively large yaw moments can be created for small yaw angles compared to the other two stiffness mechanisms, which makes the cone angle the most effective design parameter to influence the yaw stiffness.

Compared to the mechanical stiffness of a spring, the aerodynamic stiffness term does not necessarily create a restorative yaw moment. Negative force coefficient slopes over the angle of attack can create a negative stiffness term. In this case, any distur-
bance from the equilibrium point would increase the force moving the system away from the equilibrium point. An example would be the operation of the turbine during stall.

The damping mechanism for the free yaw motion is shown in Fig. 3. The aerodynamic damping of the yaw motion is created by the rotational velocity due to the yawing motion. The rotational yaw velocity is added to the wind speed on one side of the rotor and subtracted on the other side of the rotor which leads to the change in angle of attack creating an imbalance in the
loading that counteracts the yaw motion. Again, the created moment is only counteracting the yaw motion if the the slope of the airfoil coefficient over angle of attack is positive, i. e. operating in attached flow.

The stability of the equilibrium position of the yaw mode can be determined from the eigenvalue analysis of the system matrices. If the resulting real part of the eigenvalue $\lambda$ is smaller than zero and the calculated eigenfrequency $\omega$ is non-zero, there is a positively damped yaw oscillation. If the real part of the eigenvalue and the eigenfrequency are larger than zero, the yaw
equilibrium is unstable and the yaw motion is negatively damped (flutter, not to be confused with classical flutter). If the linear stiffness matrix for small yaw angles away from the equilibrium is negative definite, the system is driven away from the





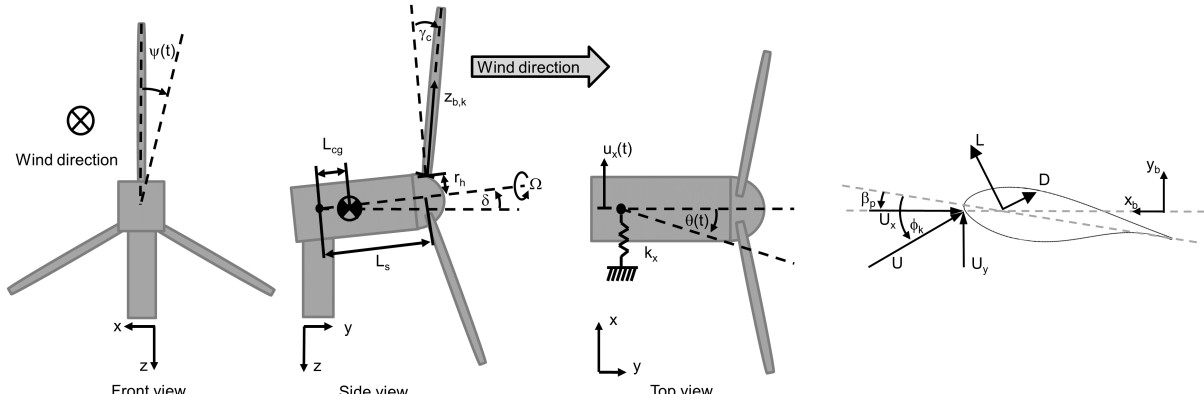

**Figure 4.** Schematics of turbine model and the according coordinate systems, front view, side view and top view (from left) and the sketch of the inflow and forces on the airfoil with the coordinate system.

equilibrium without oscillations (divergence).

Flutter instability: $\Re(\lambda) > 0$ and $|\omega| > 0$

Divergence instability: $\Re(\lambda) > 0$ and $\omega = 0$

## 2 Methods

The study is regarding two aspects. Firstly, the equilibrium yaw angle of a free yawing turbine model, which can align passively

with the wind direction, and secondly the dynamic stability of the free yaw mode. The study uses a simplified model of the Suzlon 2.1 MW turbine S111 (wind class IIIA). The original turbine has an upwind rotor with a diameter of 112 m and a tower of 90 m height. The rotor is tilted 5° and coned 3.5°. The turbine is pitch regulated at a rated wind speed of 9.5 ms$^{-1}$ and the operational range is between 4 ms$^{-1}$ and 21 ms$^{-1}$. In the investigation, the rotor configuration is changed to a downwind configuration. Thus, the rotor is shifted behind the tower, while nacelle and shaft are yawed by 180°. For the study further

simplifications are made. The blade geometry is modified: the prebend is neglected and quarter chord point of each airfoil is aligned on the pitch axis. Figure 4 shows the simplified turbine model with the geometrical parameter that will be used for a sensitivity study regarding the equilibrium yaw angle and the dynamic stability of the free yaw mode. The model is set up with two degrees of freedom (DOF) representing the free yaw motion $\theta(t)$ and the tower side-side motion $u_x(t)$ illustrated in Fig. 4. The ground fixed frame originates in the tower top center. The distance between the origin and the center of gravity of

the nacelle assembly is $L_{cg}$ and $L_s$ represents the distance from the origin to the hub center (shaft length). The hub length is $r_h$ and $z_{b,k}$ represents the position along the blade number $k$. The cone angle is denoted $\gamma_c$ and $\delta$ is the tilt angle. The azimuth position of each blade is $\psi(k,t) = \Omega t + \frac{2\pi}{3}(k-1)$, where $\Omega$ is the constant rotational speed of the shaft. The stiffness of the tower is represented by a linear spring with the stiffness $k_x$. To the right of Fig. 4, a cross section of the blade is displayed, with the inflow velocity $U$ and the respective $U_x$ and $U_y$ component, the flow angle $\phi_k$ and the pitch angle $\beta_p$ which includes




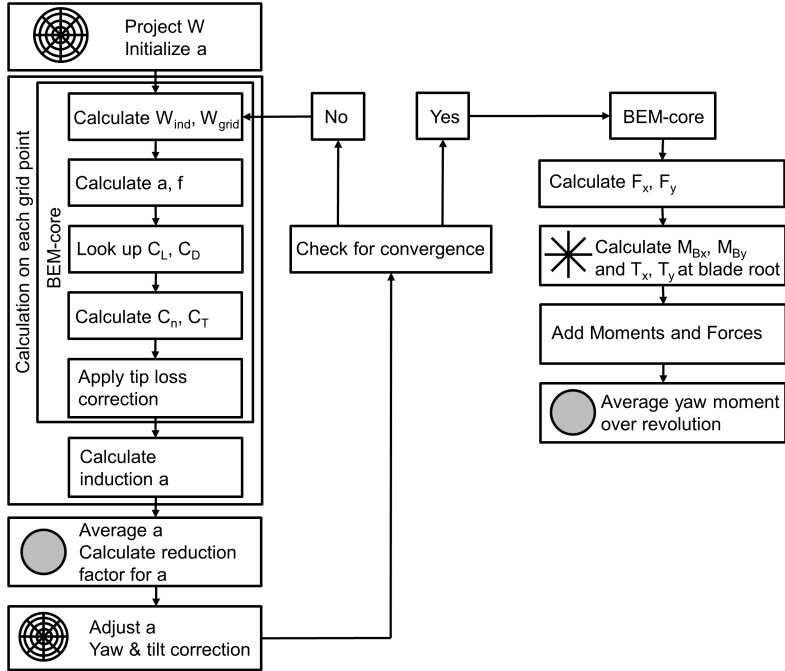

**Figure 5.** Flow chart for the implemented BEM-code to compute the equilibrium yaw angle.

the global blade pitch as well as the local twist. A steady wind field is assumed without shear, veer, inclination, turbulence or tower shadow.

### 2.1 Equilibrium yaw angle

The equilibrium yaw angle, where the aerodynamic forces are in balance, is calculated with MATLAB (Version 2018a). From a Blade Element Momentum (BEM)-code with yaw and tilt model, the forces on the rotor are calculated and the yaw angle associated with the 0-mean yaw moment on the yaw bearing is interpolated between the loading for different yaw angles. The BEM-code is based on the aerodynamic module of the aeroelastic code HAWC2 (Larsen and Hansen (2014)), as described by Madsen et al. (2018). Figure 5 shows the flow chart of the implemented BEM-method. As in HAWC2, a polar grid is set up to define the calculation points for the induction. The free wind speed $W$ is projected via a matrix rotation to the grid points, and the induction $a$ is initialized. A converging loop is set up, to calculate on each grid point induced the velocity $W_{ind}$ and the the actual velocity $W_{grid}$ on the grid points, including the blade velocity due to the rotation. From the velocity the inflow angle $\phi$ and the angle of attack $\alpha$ are calculated. The lift and drag coefficients $C_L$ and $C_D$ are interpolated within a look-up table. From this the normal force coefficient $C_n$ and the thrust coefficient $C_T$ is calculated and the tip loss correction is applied. From the corrected thrust coefficient the new induction is calculated. The values are saved for each grid point and the average induction over all grid points is calculated. From the average induction a reduction factor is calculated. This factor is applied to each grid point to reduce the average induction according to the reduced thrust from the skewed inflow. Further, the local induction





on each grid point is corrected according to the azimuth position of the blade by a yaw and a tilt factor. If the induction is then converged for all grid points within the requested tolerance, one more BEM-core operation is performed to calculate the force coefficients. From the force coefficients, the actual forces $F_x$ and $F_y$ are computed at the grid points. Those forces are integrated along the radial lines of the grid to blade root bending moments $M_{Bx}$ and $M_{By}$, as well as to shear forces at the

blade root $T_x$ and $T_y$. The total yaw moment is calculated at the hub for a full revolution, extracting values from the calculation on the grid. The total moment contribution from the out-of-plane bending moments at the hub $M_{Bx,\psi,\ hub,\ total}$ is

$$M_{Bx,\psi,\ hub,\ total} = \sum_{k=1}^{3} M_{Bx,k} \sin(\psi_k) \tag{1}$$

where $\psi_k$ is the azimuth position of the three individual blades, with $\psi = 0°$ pointing downwards. It should be noted that there is a contribution to the yaw moment from the blade root bending moments, as well as from the shear forces, which have the shaft

length as a distance to the center of yaw rotation (see Fig. 2 (c)). The total yaw moment is averaged over the rotor revolution. Finally, via interpolation the equilibrium position is found. The equilibrium yaw position is the yaw angle where the average yaw moment is zero.

For the original turbine configuration, this method is validated with a HAWC2 simulation with a free-yawing turbine model without bearing friction. Thus, the rotor can align freely with the wind field. The wind field is steady, without shear, veer,

inclination angle or tower shadow model. The dynamic stall effects are neglected. The validated BEM-code is then used for a parameter study, investigating the influence of tilt and cone angle, as well as the shaft length onto the equilibrium yaw angle of the turbine over wind speed. The operational conditions of the turbine are purely based on the free wind speed, neglecting any loss in power production due to skewed inflow.

### 2.2    Dynamic stability of the free yaw mode

To evaluate the dynamic stability of the free yaw mode, a simple 2DOF model is set up in Maple (MapleSoft, Version 2016.2). The 2DOF model based on an existing 15DOF model without cone angle, described by Hansen (2003) and Hansen (2016). The two degrees of freedom are the tower side-side motion ($u_x(t)$) and the free yaw motion ($\theta(t)$). The model is set up, without structural damping, or bearing friction. The tilt angle is assumed to be $0°$, to align the rotor with the wind direction.

The governing equations of motion are set up from the Lagrange-equation without structural damping:

$$\frac{d}{dt}\left(\frac{\partial L}{\partial \dot{x}_i}\right) - \frac{\partial L}{\partial x_i} = Q_i \text{ for } i = 1,2 \tag{2}$$

where the Lagrangian $L = T - V$ is the difference between the kinetic energy $T$ and the potential energy $V$ and $Q_i$ are the aerodynamic forces. The total kinetic energy can be written as:

$$T = \frac{1}{2}M\dot{\boldsymbol{r}}_{cg,Na}^2 + \frac{1}{2}I_z\dot{\theta}^2 + \frac{1}{2}\sum_{k=1}^{3} m_h\,\dot{\boldsymbol{r}}_{h,k} \cdot \dot{\boldsymbol{r}}_{h,k} + \frac{1}{2}\sum_{k=1}^{3}\int_0^R m_b\,\dot{\boldsymbol{r}}_{cg,k} \cdot \dot{\boldsymbol{r}}_{cg,k}\,dz \tag{3}$$

Where $M$ represents the total mass of the nacelle and shaft, $I_z$ is the total rotational inertia of the nacelle and shaft around

the yaw axis, $m_h$ is the total mass of the hub, represented as a point mass and $m_b$ is the distributed blade mass. The vectors

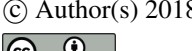



$\boldsymbol{r}_{cg,Na}$, $\boldsymbol{r}_{h,k}$ and $\boldsymbol{r}_{cg,k}$ represent the position of the nacelle mass, the hub mass and the blade center of gravity along the blade axis of the $k$-th blade with the total length $R$ and $(\dot{\ })$ denotes their respective time derivative. These position vectors can be represented as

$$\boldsymbol{r}_{cg,Na} = \begin{bmatrix} u_x - \sin(\theta) L_{cg} \\ \cos(\theta) L_{cg} \\ 0 \end{bmatrix} \tag{4}$$

,

$$\boldsymbol{r}_{h,k} = \begin{bmatrix} u_x \\ 0 \\ 0 \end{bmatrix} + \mathbf{T}_\theta \left( \begin{bmatrix} 0 \\ L_s \\ 0 \end{bmatrix} + \mathbf{T}_{\psi,k} \begin{bmatrix} 0 \\ 0 \\ r_h \end{bmatrix} \right) \tag{5}$$

and

$$\boldsymbol{r}_{cg,k} = \begin{bmatrix} u_x \\ 0 \\ 0 \end{bmatrix} + \mathbf{T}_\theta \left( \begin{bmatrix} 0 \\ L_s \\ 0 \end{bmatrix} + \mathbf{T}_{\psi,k} \left( \begin{bmatrix} 0 \\ 0 \\ r_h \end{bmatrix} + \mathbf{T}_{\gamma_c} \begin{bmatrix} 0 \\ 0 \\ z \end{bmatrix} \right) \right) \tag{6}$$

It should be noted, that the center of gravity of the blade sections is assumed to be is aligned on a straight line for simplicity. The rotation matrices for yaw $\mathbf{T}_\theta$, rotor rotation $\mathbf{T}_{\psi_k}$ and the cone angle $\mathbf{T}_{\gamma_c}$ are defined, according to the right hand rule, as

$$\mathbf{T}_\theta = \begin{bmatrix} \cos(\theta) & -\sin(\theta) & 0 \\ \sin(\theta) & \cos(\theta) & 0 \\ 0 & 0 & 1 \end{bmatrix}, \ \mathbf{T}_{\psi_k} = \begin{bmatrix} \cos(\psi_k) & 0 & \sin(\psi_k) \\ 0 & 1 & 0 \\ -\sin(\psi_k) & 0 & \cos(\psi_k) \end{bmatrix}, \ \mathbf{T}_{\gamma_c} = \begin{bmatrix} 1 & 0 & 0 \\ 0 & \cos(\gamma_c) & \sin(\gamma_c) \\ 0 & -\sin(\gamma_c) & \cos(\gamma_c) \end{bmatrix} \tag{7}$$

where the cone angle is a negative rotation for a positive cone angle.

The potential energy $V$ is formulated in the general manner, including a yaw stiffness as

$$V = \frac{1}{2}k_x u_x^2 + \frac{1}{2}G_z \theta^2 \tag{8}$$

where $G_z$ is the yaw stiffness, which will be set to $G_z = 0$ Nm$^{-1}$ for the analysis of the free yawing turbine. Inserting the Lagrangian $L$ into Eq. (2) and linearization about the equilibrium position at the steady state ($\boldsymbol{x} = \dot{\boldsymbol{x}} = 0$) gives the structural part of the linear equation of motion. It can be seen from Fig. 4 on the right that the relative inflow velocity at the blade $\boldsymbol{U}_k$, the inflow angle $\phi_k$ and the angle of attack $\alpha_k$ are

$$U_k = \sqrt{U_{x,k}^2 + U_{y,k}^2}, \ \phi_k = \arctan\left(\frac{U_{y,k}}{U_{x,k}}\right), \ \alpha_k = \phi_k - \beta_p \tag{9}$$

where $\beta_p$ includes the pitch angle and the local twist. For simplicity, it is assumed that the aerodynamic center $\boldsymbol{r}_{ac,k}$ is coinciding with the center of gravity on a straight line, the $z_b$ axis. The vector of the relative velocity is defined as

$$\boldsymbol{U} = (\mathbf{T}_\theta \mathbf{T}_{\psi_k} \mathbf{T}_{\gamma_c})^{-1} \left( \dot{\boldsymbol{r}}_{ac,k} - \begin{bmatrix} 0 \\ W \\ 0 \end{bmatrix} \right) \tag{10}$$





where $W$ is the incoming undisturbed wind to the rotor plane.

The resulting forces in the global coordinate frame can be read as

$$\boldsymbol{F_k}(z,\boldsymbol{x},\dot{\boldsymbol{x}}) = \mathbf{T}_\theta \mathbf{T}_{\psi_k} \mathbf{T}_{\gamma_c} \begin{bmatrix} f_x(z,\boldsymbol{x},\dot{\boldsymbol{x}}) \\ f_y(z,\boldsymbol{x},\dot{\boldsymbol{x}}) \\ 0 \end{bmatrix} \tag{11}$$

where the aerodynamic force components $f_x$ and $f_y$ are combined from the lift and drag coefficients as

$$f_x = \frac{1}{2}\rho c U_k^2(z,\boldsymbol{x},\dot{\boldsymbol{x}})\left(C_L(\alpha_k(z,\boldsymbol{x},\dot{\boldsymbol{x}}))\sin\ \phi_k(z,\boldsymbol{x},\dot{\boldsymbol{x}}) - C_D(\alpha_k(z,\boldsymbol{x},\dot{\boldsymbol{x}}))\cos\ \phi_k(z,\boldsymbol{x},\dot{\boldsymbol{x}})\right)$$

$$f_y = \frac{1}{2}\rho c U_k^2(z,\boldsymbol{x},\dot{\boldsymbol{x}})\left(C_L(\alpha_k(z,\boldsymbol{x},\dot{\boldsymbol{x}}))\cos\ \phi_k(z,\boldsymbol{x},\dot{\boldsymbol{x}}) + C_D(\alpha_k(z,\boldsymbol{x},\dot{\boldsymbol{x}}))\sin\ \phi_k(z,\boldsymbol{x},\dot{\boldsymbol{x}})\right) \tag{12}$$

where $\rho$ is the air density and $C_L$ and $C_D$ are the lift and drag coefficients respectively.

Inserting the time derivative of Eq. (6) representative for $\dot{\boldsymbol{r}}_{ac,k}$ and Eq. (11) and Eq. (9) to Eq. (10) and Eq. (12) and linearization around the steady state gives the linear aerodynamic matrices in the form of

$$\boldsymbol{Q} = -\mathbf{C}_{aero}\ \dot{\boldsymbol{x}} - \mathbf{K}_{aero}\ \boldsymbol{x} \tag{13}$$

where the aerodynamic forces $\boldsymbol{Q}$ have no constant component and are resulting into an aerodynamic damping matrix $\mathbf{C}_{aero}$ and an aerodynamic stiffness matrix $\mathbf{K}_{aero}$.

Here, the velocity triangle in the steady state is inserted with the components of $U_0$ as

$$U_{0x} = \Omega\,(r_h + z)\cos(\gamma_c), \ \ U_{0y} = W\cos(\gamma_c) \tag{14}$$

For simplicity in the derivation of the governing model, the induction is neglected in the upper equation. All resulting matrices
can be found in Appendix A.

From the upper equations an system matrix $A$ can be defined as

$$\mathbf{A} = \begin{bmatrix} \mathbf{0} & \mathbf{I} \\ \mathbf{M}^{-1}\,(\mathbf{K}_{struc} + \mathbf{K}_{aero}) & \mathbf{M}^{-1}\,(\mathbf{C}_{aero}) \end{bmatrix} \tag{15}$$

Where $M$ is the mass matrix, $K_{struc}$ and $K_{aero}$ are the structural and aerodynamic stiffness matrix, $C_{aero}$ is the aerodynamic damping matrix and $\mathbf{I}$ is the identity matrix. The real parts of the eigenvalues of the upper system matrix determine the damping
of the system.

A steady BEM-code without skewed inflow model is used in Matlab (R2018a), to determine the force coefficients along the blade span and to include the induction in the inflow velocity on the airfoil. The structural stiffness of the tower is tuned to account for the neglected mass distribution of the tower. Eigenanalysis of the system matrix is performed in Matlab over a range of wind speeds, and the real parts of the eigenvalue of the yaw mode are evaluated.

For the turbine configuration with the original cone, length and mass distribution, the 2DOF model is imitated in the aeroelastic

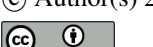



modal analysis tool HAWCStab2, described by Hansen (2004). Stiff turbine components are modelled, except the tower side-side bending and the yaw bearing is free to rotate. The real parts of the eigenvalues are compared to validate the results from the 2DOF model.

The validated model is used for a parameter study to investigate the influence of geometrical turbine parameter on the real part

5      of the yaw mode eigenvalue. The varied parameter are the cone angle, the shaft length and the position of the center of gravity of the nacelle, along the shaft.

Finally, HAWCStab2 is used to investigate, if the stability limit of the yaw mode would occur within the normal operational wind speed range of the turbine and which further degree of freedom would be needed to be included to predict any instability.

## 3   Results

10     The following section shows the results for the equilibrium yaw angle and the stability of the yaw mode are discussed.

### 3.1   Equilibrium yaw angle

Figure 6 shows the comparison of the equilibrium yaw angle found by HAWC2 and the equilibrium yaw angle found from the BEM-code, over the wind speed for the original turbine configuration with 5° tilt and 3.5° cone. The figure shows that the

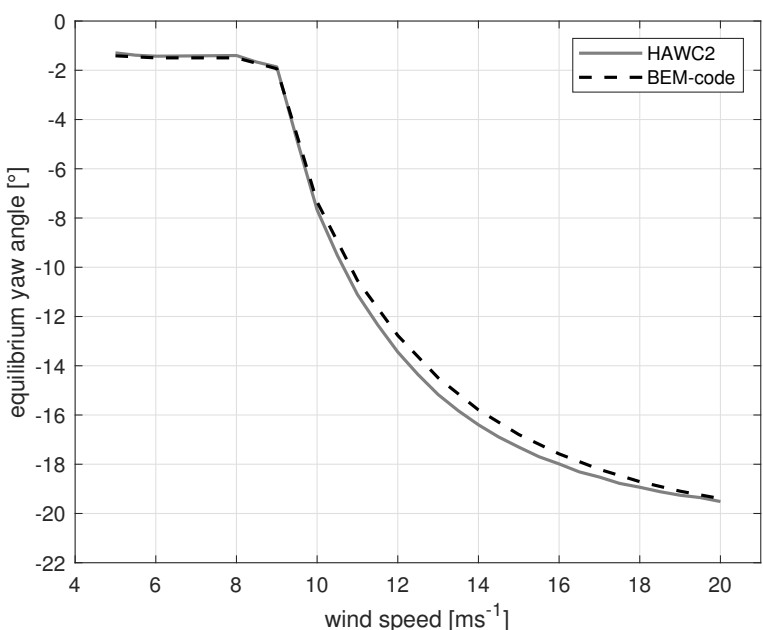

**Figure 6.** Comparison of the equilibrium yaw angle over wind speed from HAWC2 and the BEM-code for the original turbine configuration with 5° tilt and 3.5° cone.

equilibrium angle is not zero. The equilibrium yaw angle is constant at -1.4° from cut-in wind speed up to 8 ms$^{-1}$. Between

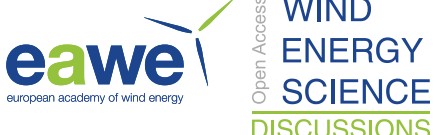

8 ms$^{-1}$ and below rated wind speed (9 ms$^{-1}$) the equilibrium yaw angle decreases slightly to -1.8°. For wind speeds higher than the rated wind speed (9.5 ms$^{-1}$), the equilibrium yaw angle decreases strongly. The slope of the equilibrium yaw angle over wind speed changes so that the equilibrium yaw angle shows a tendency to asymptotically find a minimum. The lowest observed equilibrium yaw angle of -19.4° is reached at 20 ms$^{-1}$. The equilibrium yaw angle calculated by HAWC2 and with

5 the BEM-code differ with a maximum of 0.6° at around 13 ms$^{-1}$.

The analysis shows that there will be a yaw moment even with a perfect alignment of the rotor with the wind direction, which drives the rotor to the non-zero equilibrium angle. This yaw moment is due to the tilt angle. Including a tilt angle has two effects: aerodynamically, the projection of the global wind speed leads to a yaw moment as illustrated in Fig. 1. Structurally, the tilt leads to a yaw moment as the torque axis is not perpendicular to the yaw axis and the torque $M_Q$ is projected to the

10 yaw axis with $\sin(\delta)M_Q$. While the structural effect follows the torque curve, the aerodynamic effect is influenced by the rotor speed and increases with the wind speed. When the rotor is free to align with the wind direction, the moment due to tilt pushes the rotor to a non zero yaw position. At a non zero yaw position a restorative yaw moment is present due to yaw stiffness (see Fig. 2). The rotor finds a new equilibrium yaw angle. As the equilibrium yaw angle between HAWC2 and the BEM-code agree well, the BEM-code is used for following the parameter study.

Figure 7 shows the equilibrium yaw angle (a) and the relative power production (b) in dependency on the tilt angle and wind speed. A zero tilt angle will give a zero equilibrium yaw angle, which means a full alignment of the rotor with the wind

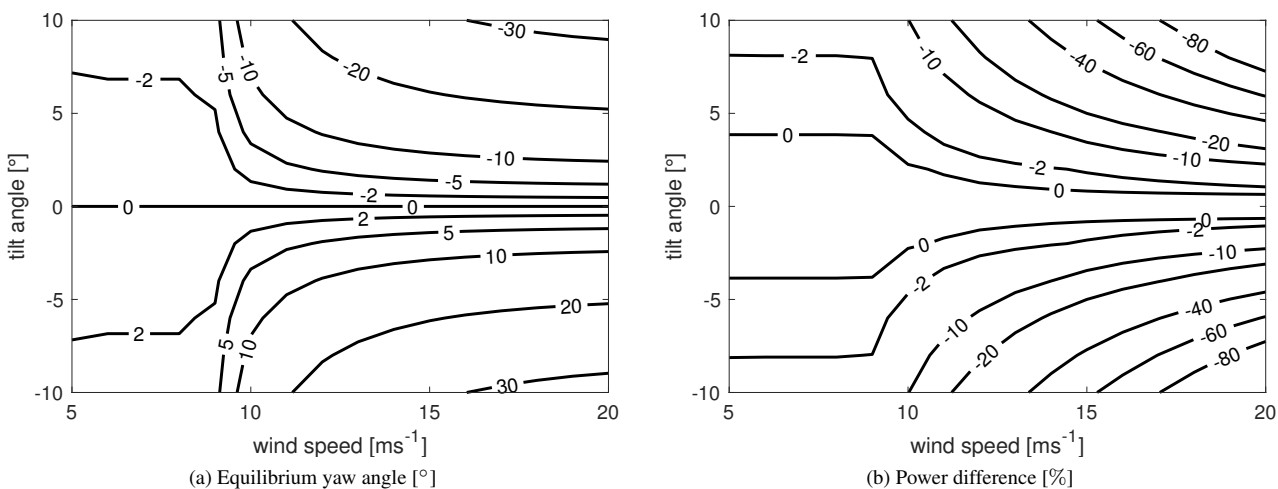

(a) Equilibrium yaw angle [°]                    (b) Power difference [%]

**Figure 7.** Equilibrium yaw angle (a) and the relative difference in power production (b) in dependency of tilt angle and wind speed variation for a turbine configuration with 3.5° cone. The relative, difference in power is compared at each calculation point relative to the power production of the original turbine, with a forced yaw alignment.

direction. Negative tilt angles show a positive equilibrium yaw angle and positive tilt angles show a negative equilibrium yaw angle. The dependency of the equilibrium yaw angle on the tilt angle is stronger for higher wind speeds. The relative power

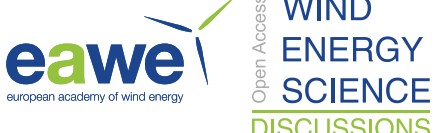



difference shows the highest losses for extreme tilt angles and high wind speeds. There is zero relative power difference at zero tilt angle.

There is no yaw moment for yaw alignment of the rotor plane with the wind direction if there is a zero tilt angle. As a yaw moment due to a tilt angle is dependent on the sine of the tilt angle, the equilibrium yaw angle is anti-symmetric around the

line of full alignment (0°). With larger tilt angles, a larger yaw moment is created aerodynamically and structurally. The larger yaw moment drives the rotor to larger equilibrium yaw angles, where a counter acting yaw moment is created from imbalance of forces by the induction variation and wind speed projections from yaw and cone angle. The power production shows the expected behaviour for a non perpendicular inflow to the rotor plane. The higher the equilibrium yaw angle, the lower the wind speed component perpendicular to the rotor plane and the lower the power production and the higher the difference to the

reference power curve.

Figure 8 shows the equilibrium yaw angle (a) and the relative difference in power production (b) for the variation of cone and wind speed. The figure had to be stitched together at the grey line, as the calculated data showed an inconsistency which could be misinterpreted as high gradients including a zero yaw angle instead of a divergence from the alignment with the wind direction. Figure 8 (a) shows that cone angles higher than 0° give a negative equilibrium yaw angle, while cone angles lower

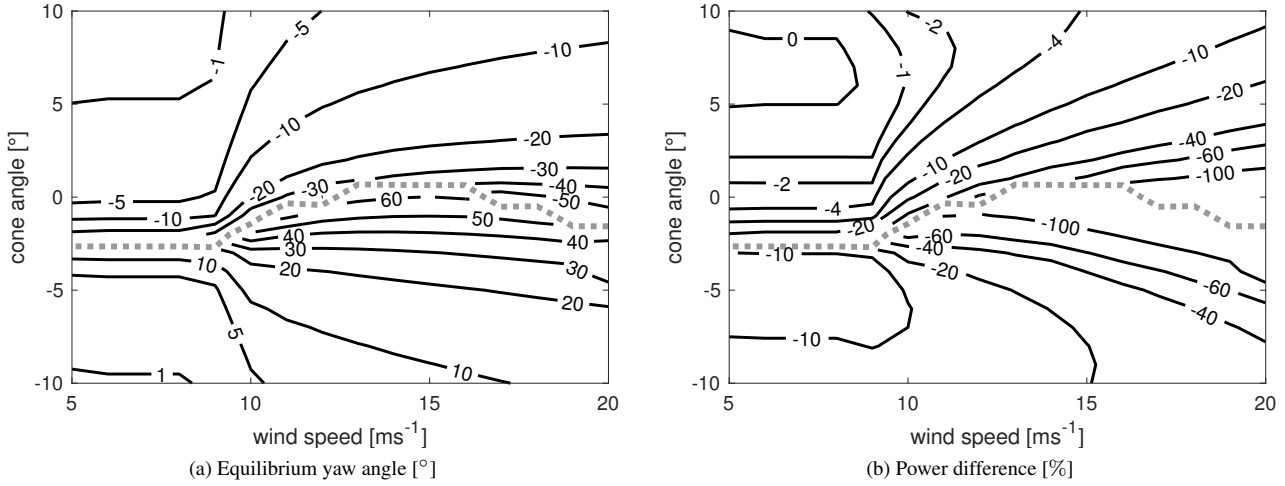

(a) Equilibrium yaw angle [°]

(b) Power difference [%]

**Figure 8.** Equilibrium yaw angle (a) and the relative difference in power production (b) in dependency of cone angle and wind speed variation for a turbine configuration with 5° tilt. The figure is stitched together from two sub-figures at the grey line.

than -2.5° give a positive equilibrium yaw angle. It can also be seen that highly positive, as well as highly negative cone angles give equilibrium yaw angles closer to zero and a smaller variation of the equilibrium yaw angle over wind speed. The higher the wind speed and the closer the wind speed to the stitching line, the larger positive or negative are the calculated equilibrium yaw angles. Figure 8 (b) shows that the extreme equilibrium yaw angles come with an extreme power loss. The negative cone angles combined with the positive equilibrium yaw angles at low wind speed are associated with a higher power loss, than the





combination of negative equilibrium yaw angles and positive cone angles.

Varying the cone angle for the tilted turbine configuration has an effect on the torque. A larger cone angle reduces the torque, which leads to a reduced projected yaw moment due to the tilt $M_{Q,\delta}$. The aerodynamic moment $M_{W,\delta}$ due to tilt on the other hand is hardly influenced. Further, larger cone angle increase the yaw moment due to yawed inflow on the coned rotor $M_{W,\theta,\gamma_c}$

(see Fig. 2 (e)). Thus, for larger positive cone angles a smaller equilibrium angle is found, not just due to the increased stiffness from cone but also due to a smaller moment from the tilt angle $M_{Q,\delta}$. For larger negative cone angles, the moment due to coned and yawed inflow is only counteracting the moments due to tilt, if the rotor is aligned with a yaw error of the opposite sign. Otherwise the stiffness from yawed and coned inflow would be negative and the force would not be restorative (see Fig. 2 (e)). As the stiffness for yawed and coned inflow is becoming very small for small cone angles, the yaw moment due to tilt

has to be balanced by the moment due to induction variation $M_a$ (see Fig. 2 (a)) and due to yawed inflow $M_{W,\theta}$ (see Fig. 2 (c)). As the two moments $M_a$ and $M_{W,\theta}$ need larger yaw angles to create a significant yaw moment, the equilibrium yaw angle becomes large for small cone angles (compare Fig. 2 (b,d,f)). Due to three dimensional effects of the wind speed projection the aerodynamic yaw moment due to tilt $M_{W,\delta}$ is not symmetric for cone angles around zero. Compared to the estimated yaw moment of the airfoil at $R = 0.75\%$ for 16 ms$^{-1}$ and 5° tilt in Fig. 1, the difference between a positive and a negative cone

angle ($\pm 0.5°$) is around 12%. As a sum the total yaw moment due to tilt is slightly different for negative and positive cone angles, the asymmetry in Fig. 8 is observed. Since the equilibrium yaw moment is not symmetric around zero, the negative cone angles combine with higher positive equilibrium yaw angles, there is a higher power loss for negative cone angles than for positive cone angles expected.

Figure 9 shows the equilibrium yaw angle (a) and the relative power difference (b) over wind speed and the shaft length factor.

Figure 9 (a) shows that the shaft length factor has nearly no influence on the equilibrium yaw position for low wind speeds. Only for high wind speeds above rated power, the equilibrium yaw angle is higher for smaller shaft length factors than for small shaft length factors. As shown on the right, the relative difference in power is hardly influenced by the shaft length factor. Only for high wind speeds less power loss is observed for higher shaft length factors than for lower shaft length factors. As discussed previously the shaft length acts as the moment arm for the summed in-plane shear forces on the hub. For the in-plane

shear forces to be significantly large a large yaw angle and high wind speeds are required to create an imbalance on the angle of attack between the upper and the lower rotor half (see Fig. 2 (c)). Only in this case the moment created from the force at the hub and the shaft length as the moment arm is large enough to counter act partly the moment created by the tilt angle. However, within the range of investigated wind speeds, the yaw misalignment with the wind direction is still so large, that hardly any power difference can be recovered by the investigated increase in shaft length.

**3.2 Dynamic stability of the free yaw mode**

The following section discusses the stability of the free yaw mode of the turbine for a tilt angle of 0° and 3.5° cone. The free yaw motion is stable around the equilibrium point, if it is positively damped, which means that the real part of the two eigenvalues for the yaw mode need to be negative. Figure 10 shows the comparison of the real parts of the eigenvalue of the yaw mode of the analytic 2DOF-model and the imitation in HAWCStab2. It can be seen at the top of the figure that there is a yaw



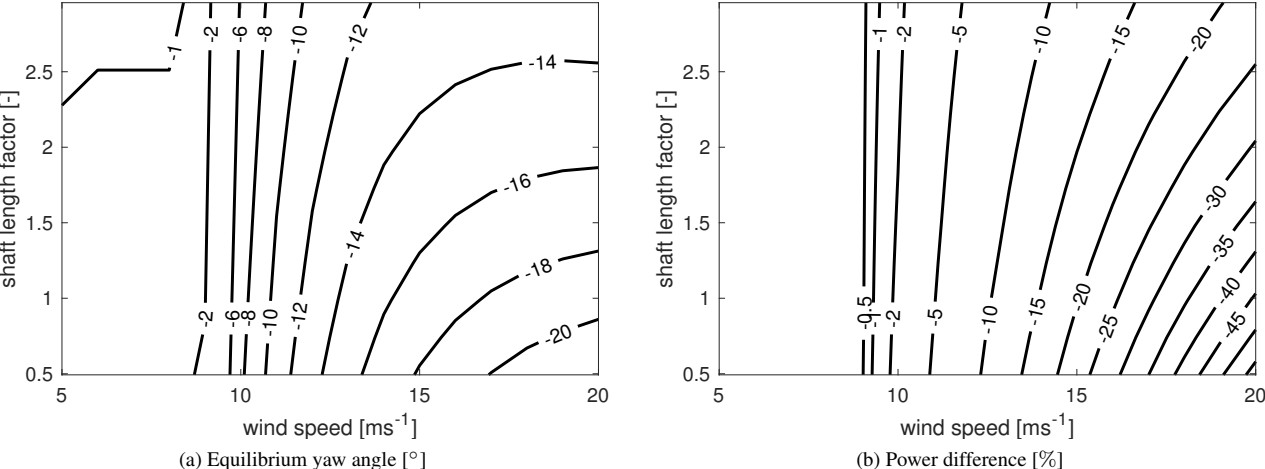

(a) Equilibrium yaw angle [°]
(b) Power difference [%]

**Figure 9.** Equilibrium yaw angle (a) and the relative difference in power production (b) in dependency of shaft length and wind speed variation for a turbine configuration with 5° tilt and 3.5° cone. The relative, difference in power is compared at each calculation point relative to the power production of the original turbine, with a forced yaw alignment.

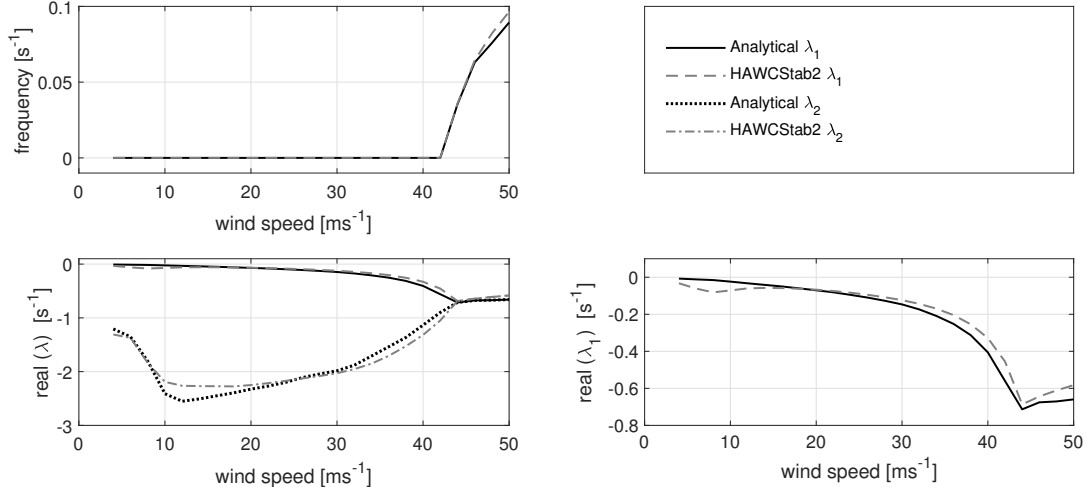

**Figure 10.** Comparison of the frequency (top left), the real part of the two eigenvalues (bottom left) and a zoom into the first eigenvalue (bottom right) of the yaw mode for the 2DOF model from the analytic solution, and the imitation in HAWCStab2.

frequency of zero up to a wind speed of $42~\mathrm{ms}^{-1}$. For higher wind speeds, the frequency is increasing up to $0.9~\mathrm{s}^{-1}$ at $50~\mathrm{ms}^{-1}$ for the solution from HAWCStab2. The results of the analytic 2DOF model and the imitation in HAWCStab2 differ maximum $0.01~\mathrm{s}^{-1}$ in the computation of the frequency of the free yaw mode. At the bottom of the figure, the real parts of the complex


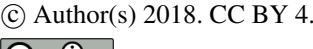


pair of eigenvalues is displayed on the left, and a zoom for the real part of the first eigenvalue is displayed on the bottom right. The first and second eigenvalue are equal only for non-zero frequencies. The first eigenvalue is generally closer to zero than the second eigenvalue for wind speeds below 44 ms$^{-1}$. The first eigenvalue decreases for wind speeds up to 8 ms$^{-1}$. The slope of the eigenvalue over wind speed changes for wind speeds above rated power. The second eigenvalue decreases to a minimum

at 10 ms$^{-1}$ (HAWCStab2) and at 12 ms$^{-1}$ (Analytical 2DOF model). For higher wind speeds the second eigenvalue increases. The total eigenvalue increases for wind speeds of 44 ms$^{-1}$ and higher. For any negative real part of the eigenvalue and zero frequency (wind speeds below 44 ms$^{-1}$), a small displacement will initiate the motion back to the equilibrium point without oscillation (convergence). For wind speeds of 44 ms$^{-1}$ and higher, there will be an oscillatory motion that will decrease in amplitude until the rotor aligns with at the equilibrium position. The difference between the solution of HAWCStab2 and from

the analytical model is up to 0.08 s$^{-1}$ for the first eigenvalue and up to 0.5 s$^{-1}$ for the second eigenvalue. The analytical 2DOF model and the imitation in HAWCStab2 cannot be expected to give the same results, since the HAWCStab2 model includes the rolling motion of the nacelle and the motion of the distributed tower mass instead of a lumped point mass. However, the real parts of the first eigenvalues of the two models are close so that the analytical model can be used for the parameter study. The results of the parameter study will be sufficient to identify the parameters that stabilize or destabilize the free yawing motion

of the turbine.

Including the aerodynamic forces to the mechanical system has two effects. Firstly, there is an aerodynamic stiffness, due to the mechanisms of wind speed projection as shown in Fig. 2. The effect of induction variation is negligible for small yaw angles. Secondly, the yaw motion creates a flow velocity that is added to the wind speed on one side of the rotor and subtracted from the wind speed on the other side of the rotor (Fig. 3), which again changes the angle of attack and therefore the aerodynamic

forces create a moment, which dampens the yaw motion.

The main influence can be observed from the slope of the lift coefficients, if the outputs from the BEM-code are manually manipulated for the eigenanalysis. As the yaw moment for moderate pitch angles is dominated by the flapwise forces, the drag and the slope of the drag coefficient are of minor influence. As the projection of the forces changes with the pitch angle, a clear dependency on the wind speed can be observed and also the change in the slope of the real part of the eigenvalue can be

observed at the rated wind speed. Further, the operational point changes so the force coefficients and their slopes are expected to change the eigenvalues over wind speed.

Figure 11 shows the real parts of the first (a) and of the second (b) eigenvalue over the variation of cone angle and wind speed. The figure shows that the real part of the first eigenvalue changes the sign and becomes positive for cone angles of -1° at 4 ms$^{-1}$ and -5° at 50 ms$^{-1}$. Thus, the zero equilibrium position becomes unstable for these negative cone angles. It can also be

seen that large positive cone angles decrease the real part of the first eigenvalue and therefore increase the damping. The larger the wind speed, the larger the effect of variation of the cone angle on the real part of the first eigenvalue. The real part of the second eigenvalue is influenced less than the real part of the first eigenvalue and varies mainly with wind speed. For very high cone angles, the minimum real part is slightly increased by 0.2 s$^{-1}$ at around 14 ms$^{-1}$. Extremely high wind speeds, larger than 40 ms$^{-1}$ show an increase of the real part of the second eigenvalue for very high cone angles. The imaginary part in the

unstable region is zero, indicating a divergence instability. In the stable region, the imaginary part of the eigenvalues is the



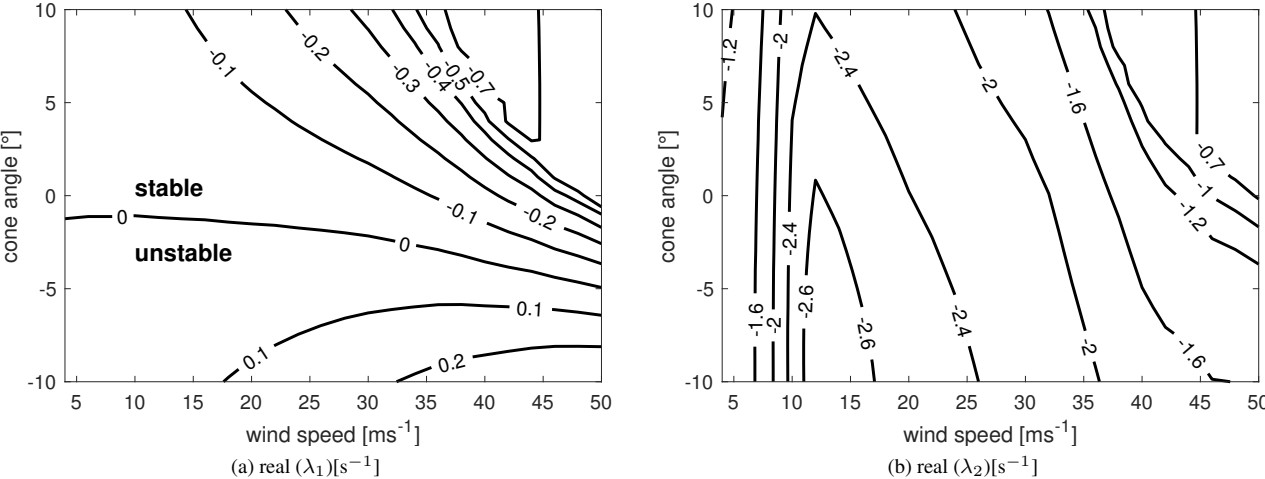

(a) real $(\lambda_1)[s^{-1}]$       (b) real $(\lambda_2)[s^{-1}]$

**Figure 11.** Real part of the first (a) and second (b) eigenvalue of the yaw mode for a variation of cone angle and wind speed from the 2DOF model.

same for high wind speeds (higher $42\ \mathrm{ms}^{-1}$) and high cone angles which means that there is a positively damped oscillatory yaw motion (flutter).

The cone angle effects mainly the aerodynamic stiffness, as shown in Fig. 2. The damping is hardly effected. However, as discussed previously, the negative cone angles can create a negative stiffness driving the system away from the equilibrium

position. A positive damping coefficient in the damping matrix cannot restore the equilibrium position in this case and the real part of the eigenvalue becomes positive. As high velocities create a positive stiffness component from the in-plane forces and due to the shaft length, the instability occurs not at zero cone angle and can tolerate slightly more negative cone angles at high wind speeds.

Figure 12 shows the real part of the first (a) and second (b) eigenvalue over a variation of wind speed and shaft length. It can

be seen that the real parts of the eigenvalues hardly change with the variation of the shaft length. A lower shaft length slightly increases the real part of the eigenvalue. High shaft length slightly decrease the minimum of the second eigenvalue at around $14\ \mathrm{ms}^{-1}$.

Large shaft length can increase the projected wind speed for the damping term, as the center of rotation is far away from the rotor plane. However, as realistic values of the shaft length are always much lower than the blade length, the influence on the

damping is very low. Also the influence on the stiffness can hardly be observed, as the effect of in plane forces is very small for small yaw angles (linearization point of $0°$ yaw). Overall, this leads to the fact that the eigenvalue of the yaw mode is hardly influenced by the shaft length within the investigated range.

A figure for the real parts of the eigenvalue changing with the position of the center of gravity is not shown. The distance of the center of gravity only effects the rotational inertia for the yaw motion. As the stiffness and damping are not effected, the

real part of the eigenvalues are hardly changing from eigenanalysis of the system matrix.





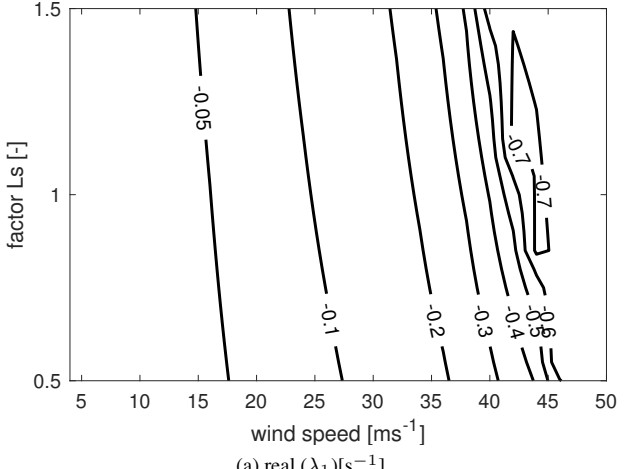
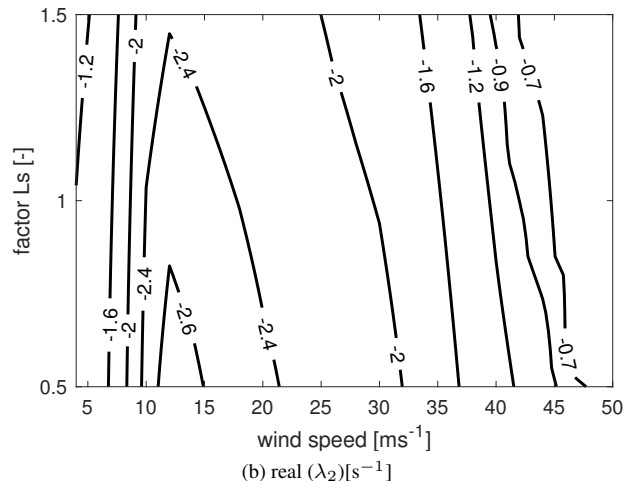

**Figure 12.** Real part of the first (a) and second (b) eigenvalue of the yaw mode for a variation of shaft length and wind speed from the 2DOF model.

Figure 13 shows the frequency at the top and at the bottom the real part of the first eigenvalue of the free yaw mode from HAWCStab2 over wind speeds. Compared are the 2DOF imitation, the extension of the 2DOF imitation with flapwise blade flexibility, the extended 2DOF model with updated steady state (deformed blade including prebend) and the full turbine model. The figure shows, that the frequency is zero for all models within the investigated wind range. The figure shows at the bottom

the characteristic behaviour of the real part of the eigenvalue of the 2DOF model imitation with HAWCStab2, already compared in Fig. 10. It can be seen that including the flapwise flexibility increases the real part of the eigenvalue significantly, especially for high wind speeds. The real part does not become positive for the investigated wind speed range due to the flapwise flexibility, as long as the steady state is not updated. The figure shows further that the real part of the eigenvalue of the yaw mode becomes positive for the 2DOF model imitation including flapwise flexibility for wind speeds of 19 ms$^{-1}$ and higher, if

the linearization is performed around the deformed steady state, including prebend (updated steady state). The real part of the eigenvalue of the free yaw mode calculated from the full turbine model differs maximum 0.005 s$^{-1}$ from the real part of the eigenvalue calculated from the extended 2DOF imitation with updated steady state.

Flapwise flexibility introduces the flapwise motion of the blades. The asymmetric flapwise motion of the forward and backward whirling mode could be stabilizing or destabilizing the yaw equilibrium, depending on the phase difference between the

yaw motion and the asymmetric flapwise motion. The phase difference between the flapwise modes an the free yaw mode is observed to be around 180°. As the flapwise motion is counter acting the yaw motion, it will decrease the damping term. Flapwise flexibility further changes the effective static cone of the system (updated steady state). Prebend and the bending of the blades towards the tower due to negative lift at high pitch angles, decrease the effective cone of the rotor. As the effective cone due to blade deflection becomes negative, a divergence instability of the zero yaw equilibrium is observed. Simulations

of time series with HAWC2 show that the turbine finds a new yaw equilibrium angle at at yaw error of around 60°. Including

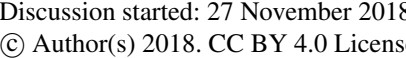





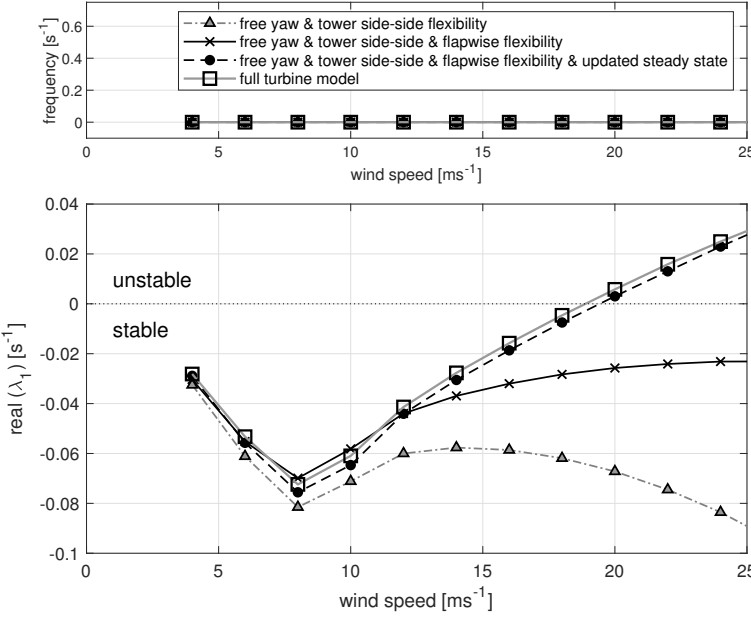

**Figure 13.** Comparison of the frequency (top) and real part of the first eigenvalue (bottom) of the yaw mode, for the imitation of the 2DOF model, the model containing additionally the blade flapwise flexibility, the model with the additional blade flapwise flexibility and the linearization around the deformed steady state and the full turbine model with a linearization around the deformed steady state and updated operational data from HAWCStab2.

the flapwise flexibility and the linearization around the deformed steady state would be sufficient to investigate the dynamics of the free yawing downwind turbine, as the difference in the real part between the full turbine model is negligible small.

## 4   Conclusions

The free yawing behaviour of the Suzlon S111 2.1 MW turbine in a downwind configuration has been investigated. A BEM-code has been used to show the equilibrium yaw angle and the parameters creating a yaw loading on the rotor. A small analytical model with only two degrees of freedom has been developed. It has been used for a fast overview and understanding the of parameters influencing the stability of the passive yaw equilibrium position, exemplified on the Suzlon turbine.

It has been seen that the original tilt angle of 5° introduces a yaw misalignment of up to -19.4° coming with a power loss of more than 20%. The tilt angle was seen to introduce a structural yaw moment from the torque projection and an aerodynamic yaw moment from the wind speed projection, as also observed by Corrigan and Viterna (1982) and Hansen (1992). Only with a tilt angle of 0° this could be fully eliminated. However, the analysis did not include any inclination angle of the wind field, which would also introduce an aerodynamic yaw moment from wind speed projection. A yaw angle due to inclination will introduce a dependency of the yaw alignment on the varying environmental conditions.

The yaw misalignment introduces a restoring yaw moment from the flapwise blade moments due to induction variation over the




rotor plane. This restoring yaw moment can be increased with an increasing cone angle, as the combination of cone and yaw angle creates a favorable wind speed projection and therefore increases the yaw stiffness. This result confirms the observations of the measurements from for example Verelst and Larsen (2010) and Kress et al. (2015).

With a significantly large yaw angle, the wind speed projection leads to a in-plane force imbalance that increases the restoring

yaw moment. In conclusion, an in-plane force due to load imbalance will also be created from the tower shadow and wind shear. In contrast to the previous effect, this force imbalance will also exist when the rotor is fully aligned with the wind direction and it will vary with varying wind conditions. Such a negative effect from vertical wind shear and tower shadow has already been observed for example by Hansen (1992).

The eigenanalysis of a 2DOF model of the turbine with 0° has been conducted. It has been observed that the cone angle can

significantly increase the real part of the eigenvalue of the yaw mode and therefore stabilize the yaw equilibrium as it increases a positive stiffness term. It has further been observed that a too small cone angles can give a negative stiffness term and therefore leads to a positive real part of the eigenvalue and an instability in the yaw mode.

Modelling the free yawing motion with 2DOF has been seen to be not sufficient, as flapwise blade motion changes the stiffness and the damping of the free yaw mode. The comparison with HAWCStab2 showed that flapwise blade flexibility significantly

increases the real part of the eigenvalue and destabilized the yaw equilibrium. The phase difference between yaw and asymmetric flapwise blade mode decreases significantly the damping of the free yaw mode. The stiffness is mainly influenced by flapwise blade deformation as the steady state blade deflection decreases the effective cone angle.

Over all, this analysis showed clearly that the S111 turbine in downwind configuration will not align with the wind direction and the power loss is significant. Further, changing wind conditions such as inclination angle or wind shear will lead to a yaw

misalignment that will change with the environmental conditions. As the free yaw mode further becomes unstable for high wind speeds, it will not be possible to run the S111 in a free yawing downwind configuration. Stabilizing the free yaw mode and increasing the alignment with high cone angles might be possible. However, as the yaw drives will always be needed as a cable unwinding mechanism, it is highly doubtful that the passively free yawing downwind turbine can be a cost efficient solution.

**Appendix A: Matrices for the 2DOF model**

**Structural Matrices**

The mass matrix results into

$$\mathbf{M} = \begin{bmatrix} 3\left(\int_0^R m_b(z)dz\right) + 3m_h + M & M_{12} \\ M_{21} & M_{22} \end{bmatrix} \tag{A1}$$

where the coupling term between the tower side-side motion and the nacelle yaw are

$$M_{12} = M_{21} = -3\left(\int_0^R m_b(z)(L_s + \sin(\gamma_c)z)dz\right) - 3m_hL_s - ML_{cg} \tag{A2}$$





and the mass element for the yaw motion is

$$M_{22} = \frac{3}{2} \int_0^R m_b(z) \left( -\cos(\gamma_c)^2 z^2 + 4L_s \sin(\gamma_c)z + 2r_h \cos(\gamma_c)z + 2L_s^2 + rh^2 + 2z^2 \right) dz$$

$$+ \frac{3}{2} m_h r_h^2 + 3L_s^2 m_h + M L_{cg}^2 + I_z \tag{A3}$$

The resulting stiffness matrix is

$$\mathbf{K} = \begin{bmatrix} k_x & 0 \\ 0 & G_z \end{bmatrix} \tag{A4}$$

In the stiffness matrix the spring stiffness can be found on the diagonal, while there is no coupling from the stiffness in the off diagonal elements.

The mass matrix on the other hand is fully populated. On the first element is the total mass of the turbine that will be moved with the tower side-side motion. On the second diagonal element there is the mass moment of inertia for the rotation around the yaw center. This includes the mass moment of inertia of blades, hub and nacelle-shaft assembly, as well as their respective

Steiner-radius to the center of rotation. The coupling terms on the off-diagonal the mass elements times the respective radius to the rotational axis.

**Aerodynamic Matrices**

The resulting stiffness matrix $\mathbf{K}_{aero}$ is only populated in the second collumn with a coupling term from the tower side-side motion and an aerodynamic stiffness term for the yaw motion.

$$\mathbf{K}_{aero} = \frac{1}{4} c\rho W^2 \int_0^R \begin{bmatrix} 0 & K_{12,aero} \\ 0 & K_{22,aero} \end{bmatrix} dz \tag{A5}$$

The coupling coefficient $K_{12,aero}$ from the tower motion to yaw motion is

$$K_{12,aero} = 12 C_{y0} \cos(\gamma_c)^3 + 3\lambda \left( C'_{y0} - C_{x0} \right) \cos(\gamma_c)^2 + 3 \left( 2\lambda^2 C_{y0} - C'_{x0} - 3C_{y0} \right) \cos(\gamma_c) + 3\lambda \left( 3C_{x0} - C'_{y0} \right) \tag{A6}$$

and the aerodynamic yaw coefficient is

$$K_{22,aero} = -6 L_s C_{y0} \cos(\gamma_c)^3$$
$$+ \left[ 6 r_h C_{y0} \sin(\gamma_c) + 3\lambda L_s \left( C_{x0} - C'_{y0} \right) \right] \cos(\gamma_c)^2$$
$$+ \left[ \left( 3\lambda r_h \left( C'_{y0} - C_{x0} \right) + 3z \left( 3C_{y0} + C'_{x0} \right) \right) \sin(\gamma_c) + 3L_s \left( 3C_{y0} + C'_{x0} \right) \right] \cos(\gamma_c)$$
$$- 3\lambda \left( L_s + sin(\gamma_c)z \right) \left( 3C_{x0} - C'_{y0} \right) \tag{A7}$$

The aerodynamic damping matrix $\mathbf{C}_{aero}$ is symmetric and fully populated.

$$\mathbf{C}_{aero} = \frac{1}{8} c\rho W \begin{bmatrix} C_{11,aero} & C_{12,aero} \\ C_{21,aero} & C_{22,aero} \end{bmatrix} \tag{A8}$$




with the aerodynamic tower side-side coefficient

$$C_{11,aero} = -12C_{y0}\cos(\gamma_c)^3 + 6\lambda\left(C_{x0} - C'_{y0}\right)\cos(\gamma_c)^2 + 6\left(C'_{x0} + 3C_{y0}\right)\cos(\gamma_c) + 6\lambda\left(C'_{y0} - 3C_{x0}\right)$$ (A9)

The aerodynamic coupling coefficients $C_{12,aero} = C_{21,aero}$ wich is $C_{21,aero} = -2K_{22,aero}$

$$
\begin{aligned}
C_{12,aero} = {}& 12L_sC_{y0}\cos(\gamma_c)^3 \\
&+ \left[-12r_hC_{y0}\sin(\gamma_c) - 6\lambda L_s\left(C_{x0} - C'_{y0}\right)\right]\cos(\gamma_c)^2 \\
&+ \left[\left(6\lambda r_h\left(C_{x0} - C'_{y0}\right) - 6z\left(3C_{y0} + C'_{x0}\right)\right)\sin(\gamma_c) - 6L_s\left(3C_{y0} + C'_{x0}\right)\right]\cos(\gamma_c) \\
&+ 6\lambda\left(L_s + z\sin(\gamma_c)\right)\left(3C_{x0} - C'_{y0}\right)
\end{aligned}
$$ (A10)

The aerodynamic damping coefficient of the yaw motion is

$$
\begin{aligned}
C_{22,aero} = {}& \left(12\left(r_h^2 - L_s^2\right)C_{y0} - 6z^2\left(C'_{x0} + C_{y0}\right)\right)\cos(\gamma_c)^3 \\
&+ \left[24L_sr_h\sin(\gamma_c)C_{y0} + \left(6L_s^2\left(C_{x0} - C'_{y0}\right) + 12z^2C_{x0} - 6\lambda r_h^2\left(C_{x0} - C_{y0}\right)\right) + 24r_hzC_{y0}\right]cos(\gamma_c)^2 \\
&+ \left[12L_s\left(z\left(3C_{y0} + C'_{x0}\right) - \lambda r_h\left(C_{x0} - C'_{y0}\right)\right)\sin(\gamma_c) - 12\lambda r_hz\left(C_{x0} - C'_{y0}\right) + \left(6\left(L_s^2 + z^2\right)\left(3C_{y0} + C'_{x0}\right)\right)\right]\cos(\gamma_c) \\
&- 18\lambda\left(L_s^2 + 2L_sz\sin(\gamma_c) + z^2\right)\left(C_{x0} - C'_{y0}\right)
\end{aligned}
$$

(A11)

Where the subscript 0 indicates the steady state values. The substitutes in the matrix coefficient have the following definitions for the tangential $C_{x0}$ and the normal force $C_{y0}$ coefficient

$$C_{x0} = C_{L0}\sin(\phi_0) - C_{D0}\cos(\phi_0)$$ (A12)

and

$$C_{y0} = C_{L0}\cos(\phi_0) + C_{D0}\sin(\phi_0)$$ (A13)

derivatives of the force coefficients over alpha denoted b ′ are stated as

$$C'_{x0} = C'_{L0}\sin(\phi_0) - C'_{D0}\cos(\phi_0)$$ (A14)

and

$$C'_{y0} = C'_{L0}\cos(\phi_0) + C'_{D0}\sin(\phi_0)$$ (A15)



*Author contributions.* Gesine Wanke and Morten H. Hansen set up the 2DOF model and validated the model. Gesine Wanke and Torben J. Larsen have set up and validated the BEM-code to calculate equilibrium yaw angles. Gesine Wanke carried out the calculations. All authors have interpreted the obtained data. Gesine Wanke has prepared the manuscript with revisions of all co-authors.

*Competing interests.* This Project is an industrial PhD project founded by the Innovation Fund Denmark and Suzlons Blade Science Center.

5 Gesine Wanke is employed at Suzlons Blade Science Center.



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
