# Peer review of "Qualitative yaw stability analysis of free-yawing downwind turbines"

_Wind Energy Science, 2018_

## Referee Comment (RC1) · Anonymous Referee #1 · 20 Dec 2018

WES-2018-68 review

General comments: A good project investigating design parameters for free-yaw behavior of a downwind turbine, with verification of lower-order dynamic models comparing well with high-fidelity modeling tools.

Specific comments: 1. Should state somewhere that the turbine is 3-bladed and describe the regulation type: variable/constant speed below rated then pitch/stall regulation for power/rotor speed. 2. Also state if the rotor shaft intersects (no offset) the yaw axis. 3. Suggest nomenclature list 4. I don't think that Madsen 2018 should be referenced, unless it is accepted for publication prior to this paper. 5. Recommend checking "Wind Turbine Engineering Design" by Eggleston and Stoddard about yaw behavior. 6. Page 21 line 23- not clear on what is the cost-effective solution- going

downwind? I would speculate that the yaw system required for downwind alignment and de-wrapping (and associated bearings, tower material thickness) would have a much lower torque requirement than that required for upwind operation. 7. Note that I did not review the equations in the appendix in detail. Recommend that the author double-check these.

Technical corrections: Page 1 line 15: change to "loaded higher" P(age)1 L(ine)21-define "turbine" tilt- do you mean shaft? P2 L33- Sine to Since P4 L15-18- these two sentences are not clear P4 L21- change "as it will" to "as flexibility will" P4 L25- not clear if this a vector or scalar quantity. For either, explain/show (another figure) what axis it is about. Figure 1 (and others) the vector triangle on the left/right is not clear- do the grey lines mean something? What is the resultant vector? P6 L18- change "smaller" to "less" Figure 4 increase font size P7 L11- explain the "geometrical parameter" - I do not see it in Figure 4 P9 L13- state if tower shadow modeling is included P9 L22- change sentence to "The model does not include structural damping or bearing friction." P11 L10 change "are resulting into an" to "result in the" P11 L11 change "an" to "the" P11 L12 Sentence not clear, perhaps include figure? P11 L14 not clear on upper equation; reference number P11 L18 Bold M P11 L19 reference matrix number P12 L6 remove comma P12 L6 remove comma after "investigate" P12 L6 remove "to be included" and "any" P13 L14 change to "well; the BEM-code is therefore used for the parameter study" P14 L9 change to "...rotor plane, the lower power production, and the higher..." P14 L12 sentence note clear P15 L18 should "expected" be removed? P15 L19 define/show equation for the shaft length factor P15 L33 change "need to be" to "are" P21 L9 "with 0 degrees" not clear P22 L13 spelling "column" P24 L4 spelling "funded" P25 L4 caps for MOD-0

---

## Referee Comment (RC2) · Riziotis (Referee) · 14 Jan 2019

The paper presents a reduced order model for the aeroelastic analysis of downwind wind turbine configurations. Parametric analysis of the stability of the system and the equilibrium yaw position is performed with parameters the tilt and cone angle and the shaft length. The results of the reduced order model are compared against simulation results from more advanced tools that simulate the full dynamics of the wind turbine system. It is a well written paper and a nice qualitative analysis of the downwind free yaw concept with in depth explanations of the underlying physical mechanisms. A few points that the authors should consider/comment are the following: 1) One critical choice of the work is that the WT system is simulated as a 2 dof system. The only dof, additional to the yaw dof, considered in the analysis is the lateral deflection of the

tower. In my opinion the rationale of the above choice should be explained in section 2. It could be justified in relation to the full aeroelastic stability analysis performed with HAWCStab2. For example no other critical mode than the lateral bending mode appears in the HAWCStab2 analysis under any circumstances? 2) Several different models have been employed in the present work. Perhaps a table listing the main and most important features of the above models, per model, could be included. For example, which of the models include yaw skewness effect or unsteady aerodynamic and dynamic inflow effects etc? Sometimes it becomes a bit confusing for the reader who is not familiar with all the above modelling options to follow the comparisons.

Other specific comments and editorial can be found in the attached pdf.

Please also note the supplement to this comment:
https://www.wind-energ-sci-discuss.net/wes-2018-69/wes-2018-69-RC2-supplement.pdf

[Figure]

**Supplement:**

[revised manuscript text omitted]

---

## Author Response (AR1)

**Answer to Referee comment #1**

Thanks a lot for your effort and the very helpful feedback on our work.

Answer to the specific comments:

1. Turbine blade number and regulation
   The turbine is three-bladed and originally with a variable speed generator and pitch regulation, with a constant power control strategy. However, the controller is not implemented and for all shown calculation and simulations a prescribed rotor speed and pitch angle are used according to the wind speed.
2. Rotor shaft intersection
   The rotor shaft intersects with the yaw axis and there is no offset between the shaft and the yaw axis.
3. Nomenclature list
   Nomenclature list can be implemented for a new article version.
4. Citation of Madsen 2018
   The article is not published yet, however, we would prefer to cite their work if it is published by the time that a final version of this article is published. The full BEM-code implementation is heavily based on their work and is a major reason for the relative good agreement between the simple BEM and the HAWC2 results. However, in case that it is not published by the time we submit a final version of the article, we will consider to add their work to the acknowledgements instead.
5. "Wind Turbine Engineering Design" by Eggleston and Stoddard
   Thanks for the additional reference, it will be checked.
6. Cost-effective solution
   It is true, that the yaw torque during normal operation will be lower for the free yawing downwind configuration. However, a full alignment with the wind direction cannot be guaranteed. Even without a tilt angle for the shaft, the tower fore-aft bending will result into such a tilt angle and therefore yaw misalignment. A misaligning effect can also be expected from any horizontal wind gradient or wind shear. A yaw misalignment will result into a significant power loss and therefore a decreased AEP. Considering the full turbine costs in terms of levelized cost of energy (LCOE) the whole yaw system has a very small impact. Even if a significant saving on the associated bearings could be achieved the impact on the LCOE will be very low. The loss in power due to misalignment, and the associated cost on the AEP on the other hand will be relatively high.
   The tower wall thickness is designed for maximum bending moment resulting from load cases with a turbine shut down and there are no cost savings associated with a change in the yaw concept.
   This study does not allow a conclusion if a downwind concept in general would be beneficial. However, there is no major cost advantage expected comparing a free-yawing downwind concept to a controlled yawing concept.
7. Equations in the appendix
   Authors will double-check these for a new article version

Answer to technical corrections (for questions only, requests of change will be implemented in a new article version):

P(age)1 L(ine)21 –
turbine tilt means the tilt angle of the shaft

Figure 1 –grey lines in the vector triangles
the grey lines in the vector triangles are supposed to show the original vector triangle as if the described effect was not present. It will be considered to delete the grey lines for clearer figures in a new version.

P11 L12 steady state velocity triangle
The steady state is describing the velocity triangle of Figure 4. This will be stated clearer to avoid confusion. Tis should also clarify the comment for the next line: the derivation is based on the inflow triangle, which does not include the induced winds on the rotor. These are associated later to the triangle, when the matrices are calculated in Matlab.

**Answer to Referee comment #2**

Thank you for your effort and the helpful feedback on our work.

1. A 2DOF model is chosen, in the attempt to keep the model as simple and fast as possible. The advantage would be, that such model could in principle be used to make basic design choices very fast. The tower side-side is chosen as the second degree of freedom, as it couples directly to the yaw motion via the shaft length and the rotor mass. However, as seen from the results this is not sufficient, as the flapwise blade motion also has a large influence.

2. (& PDF-comment) A clearer naming of the different BEM-methods will be introduced for more clarification. The description cannot be removed, since the paper of Madsen 2018 has not been submitted yet and it was pointed out in the previous reviewer comment, that it is not citable.

3. (PDF-comment) The rotor in Figure 1 and 2 should not be counterclockwise rotating. Velocity triangles need to be corrected in order to resemble the computations.

4. (PDF-comment) Estimation of the equilibrium position from 0-mean yaw moment. It is true, that loads due to acceleration could lead to a different yaw equilibrium from the averaged equilibrium from the BEM-code. However, the HAWC2 simulation is with extremely slow increase in wind speed to avoid such effect. In the HAWC2 simulations no yaw oscillations have been observed to prove such significant loads due to accelerations.

**List of changes according to Referee comment #1**

Changes for the specific comments:

1. Turbine blade number and regulation
   Added in 2 Methods
2. Rotor shaft intersection
   Added in 2 Methods
3. Nomenclature list
   Nomenclature list is implemented.
4. Citation of Madsen 2018
   The article has not been published by the time of submission of the revised version of the article. It has been substituted by an article describing the underlying principles for the implementation of the skewed wake model in 2 Methods.
5. "Wind Turbine Engineering Design" by Eggleston and Stoddard
   An additional paragraph has been added.
6. Cost-effective solution
   The term "cost efficient" has been specified in order to clarify the comparison of a downwind turbine with or without free yaw control.
7. Equations in the appendix
   Authors have double-checked these.

Changes for technical corrections:

P(age)1 L(ine)21 –
turbine tilt has been changed to shaft tilt.

P3L15-18 – sentences have been changed in introduction last paragraph.

Figure 1 (and others)–grey lines in the vector triangles
Figures have been updated and also needed correction according to referee comments #2.

Figure 4 – font size has been increased

P7 L11 geometrical parameter are listed in 2 Method (marked version P8 L11)

P9 L13 tower shadow model is not included as stated in wind field description in the end of 2 Methods.

P11 L12 - referenced to the Figure 4 for description

P11 L 14 – Equation number added

P14 L12 sentence changed to "Here the angles tend to increase to very large positive and negative angles, rather than decrease to zero as a continuous figure would suggest."

P21L9 – "0°" changed to "without tilt angle"

All other comments regarding wording and spelling have been implemented according to suggestion

**Changes according to Referee comment #2**

Changes according to direct comments

    1.) Added paragraph in section 2.2

"A 2DOF model is chosen, in the attempt to keep the model as simple and fast as possible. The advantage would be, that such model could in principle be used to make basic design choices very fast. The tower side-side is chosen as the second degree of freedom, as it couples directly to the yaw motion via the shaft length and the rotor mass. "

"Finally, HAWCStab2 is used to investigate if the stability limit of the yaw mode would occur within the normal operational wind speed range of the turbine and which further degree of freedom would be needed to predict instability."

Changed to (at end of section 2.2)

"Finally, HAWCStab2 is used to investigate if the stability limit of the yaw mode would occur within the normal operational wind speed range of the turbine and which further degree of freedom, additional to the tower side-side and yaw, would be needed to predict instability."

    2.) Confusion from models: all models have been given a specific naming within the description and the naming is used throughout the manuscript for clear identification.

Changes according to the comments in supplement

P3 L27 - No further change as all effects are regarded separately

Figure 1 (and describing text (p.4)) - Rotation direction was the same, however the Figure (and the describing text) was sketched wrong and had to be corrected.
For sign definition reference to Figure 4 is added and text changed according to suggestion. (see also comment on Figure 4)

P8 L3 – BEM description could not be deleted as the reference turned out to be not citable

P8 L6 – added "assuming that the effect of inertial terms is negligible" since these were not observed in time simulations.

P8 L10- rephrased to "Within a converging loop the induced velocity and the actual velocity at each grid point are calculated".

P10 L15 – details on steady state of x=0 added "The linearization around a steady state of x=0 assumes that there exists and equilibrium position where the rotor is fully aligned with the wind direction, as the tilt angle is 0°. From the linearized model the stability due to small angle variations around the equilibrium can be investigated.

P11 L21 & P12 L13 -  BEM-code is specified

P13 Figure 7 – there is no controller update according to power loss associated with wind direction misalignment. This is stated in the end of section 2.1

P14 Figure 8 - angle definition has been added with Figure 4 as suggested

Suggested changes in wording and spelling have been implemented.

[revised manuscript text omitted]